# Homocysteine Plasmatic Concentration in Brain-Injured Neurocritical Care Patients: Systematic Review of Clinical Evidence

**DOI:** 10.3390/jcm11020394

**Published:** 2022-01-13

**Authors:** Maria Paola Lauretta, Rita Maria Melotti, Corinne Sangermano, Anneliya Maria George, Rafael Badenes, Federico Bilotta

**Affiliations:** 1Department of Anaesthesia and Pain Management, IRCCS Policlinico S. Orsola-Malpighi of Bologna, University of Bologna, 40138 Bologna, Italy; mariapaola.lauretta@aosp.bo.it; 2Department of Anaesthesia and Pain Management, IRCCS Policlinico S. Orsola-Malpighi of Bologna, Alma Mater Studiorum, University of Bologna, 40138 Bologna, Italy; ritamaria.melotti@aosp.bo.it; 3Department of Anaesthesia, Intensive Care and Pain Management, Umberto I Hospital, Sapienza University of Rome, 00161 Rome, Italy; corinne.sangermano@gmail.com (C.S.); anneliya.george16201@gmail.com (A.M.G.); federico.bilotta@uniroma1.it (F.B.); 4Department of Anesthesiology and Surgical Trauma Intensive Care, Hospital Clínic Universitar de Valencia, University of Valencia, 46010 Valencia, Spain

**Keywords:** homocysteine, cerebrovascular disease, perioperative care

## Abstract

Background: Hyperhomocysteinemia (HHcy) is considered as an independent risk factor for several diseases, such as cardiovascular, neurological and autoimmune conditions. Atherothrombotic events, as a result of endothelial dysfunction and increased inflammation, are the main mechanisms involved in vascular damage. This review article reports clinical evidence on the relationship between the concentration of plasmatic homocysteine (Hcy) and acute brain injury (ABI) in neurocritical care patients. Materials and methods: a systematic search of articles in the PubMed and EMBASE databases was conducted, of which only complete studies, published in English in peer-reviewed journals, were included. Results: A total of 33 articles, which can be divided into the following 3 subchapters, are present: homocysteine and acute ischemic stroke (AIS); homocysteine and traumatic brain injury (TBI); homocysteine and intracranial hemorrhage (ICH)/subarachnoid hemorrhage (SAH). This confirms that HHcy is an independent risk factor for ABI and a marker of poor prognosis in the case of stroke, ICH, SAH and TBI. Conclusions: Several studies elucidate that Hcy levels influence the patient’s prognosis in ABI and, in some cases, the risk of recurrence. Hcy appears as biochemical marker that can be used by neuro-intensivists as an indicator for risk stratification. Moreover, a nutraceutical approach, including folic acid, the vitamins B6 and B12, reduces the risk of thrombosis, cardiovascular and neurological dysfunction in patients with severe HHcy that were admitted for neurocritical care.

## 1. Introduction

Hyperhomocysteinemia (HHcy) is an independent risk factor for cardiovascular, neurological and autoimmune diseases [1]. Homocysteine (Hcy) is a key metabolite involved in the biosynthesis and metabolism of methionine (Met). Although Hcy is not directly involved in protein synthesis, its role in folate and choline catabolism is fundamental in the regulation of Met activity and for the synthesis of several proteins. Hcy could be used to donate methyl groups, which is the base for the synthesis of methylated compounds, while inorganic sulfate is important for the synthesis of sulfur-containing amino acids (Figure 1) [2]. Blood sampling can be used to measure Hcy plasmatic levels. They can significantly vary individually, as follows: physiological levels are between 5 and 10 μmol/L, and levels above 10 μmol/L result in the condition of HHcy. HHcy is classified as “mild” if the levels range from 15 to 30 μmol/L, “intermediate” if they range from 30 to 100 μmol/L, and “severe” for values over 100 μmol/L [3]. Persistent HHcy is an established biomarker of increased cardiovascular risk in out-of-hospital patients; it promotes atherothrombotic events and increases inflammation, leading to vessel damage [4]. The main mechanism of cardiovascular risk by HHcy is pervasive endothelial dysfunction, engaging both small and large vessels. Both the reduction in endothelial vasodilator nitrogen monoxide (NO) and the increase in oxidative stress, due to the production of reactive oxygen species (ROS), are involved in this pathway [5]. Since ROS easily interacts with NO to form peroxynitrite (ONOO^−^), an aggressive oxidizing reactive molecule, thromboxane A2 (TxA2) synthesis, with subsequent arteriolar vasoconstriction, is raised [6]. Furthermore, Hcy promotes atherosclerotic lesions, activating the transcription of nuclear factor kappa-light-chain-enhancer of activated B cells (NF-κB), which increases the endothelial expression of monocyte chemoattractant protein-1 (MCP-1) and inflammatory cytokine production, such as interleukin-8 (IL-8) (Figure 2) [7].

Recent evidence suggests that Hcy worsens acute brain injury (ABI) and other neurological conditions, such as post-stroke early neurological dementia (END), brain atrophy, Parkinson & Alzheimer disease and epilepsy, although its molecular mechanism in this role is not fully clarified [8]. First of all, ROS, in addition to its cardiovascular effects, negatively influences the brain, with the development of neuronal damage, and leads to neuronal cell death and brain atrophy. An increased ROS concentration promotes mitochondrial dysfunction through copper (Cu^2+^) chelation, which results in cytochrome C oxidase inactivation; cytochrome C is an enzyme for the ATP synthesis in the respiratory electron transport chain [9]. Moreover, Hcy is an amino acid with excitatory activity on N-methyl-D-aspartate glutamate receptors (NMDA), leading to a toxic condition called “*excitotoxicity*”, which is responsible for changes in the homeostasis of intracellular calcium (Ca^2+^) and the development of numerous neurological conditions [10]. Other pathways that associate HHcy effects to cell damage in the nervous system are related to the inflammatory response: Hcy is able to increase NF-kB expression and its activity in neuronal cells, as well as many pro-apoptotic markers, such as Bax, p53 and caspase-3 (Figure 3) [11].

Despite several studies suggesting that the plasmatic concentration of Hcy might serve as a predictor for cardiovascular and cognitive complications, especially in patients with ABI, the available evidence of its predictive role in neurocritical care is sparse and there is no literature that specifically presents the possible relationship.

The aim of this systematic review (SR) is to report clinical evidence on the role of the plasmatic concentration of Hyc in brain-injured neurocritical care patients.

## 2. Materials and Methods

The methodological features of this SR have been registered and accepted into the international prospective register of systematic review database (PROSPERO registration number: CRD42021254134). The reporting of this SR was guided by the standards of the Preferred Reporting Items for Systematic Review and Meta-Analysis (PRISMA) statement [12]. Two medical databases, PubMed and MEDLINE, were queried to identify published literature for inclusion in this SR. Terms and the research strategy employed for the literature search are listed in Appendix A. Only studies published as full text in peer-reviewed journals were considered eligible for this SR. Two authors (MPL and CS) independently screened retrieved studies and assessed the titles, abstracts, and the full-text articles. Clinical studies, in adult patients (>18 years), published in English, that reported original information on the role of Hcy in ABI and perioperative neurocritical care were included. Reviews and editorials (i.e., articles not presenting original clinical research) were excluded. After hand searching and revision of the full text, duplicates were eliminated by Zotero tool. A standardized data extraction form was used by the authors to examine and compare results from relevant studies [13].

The risk of bias in the included studies was assessed according to Cochrane Bias Methods Group guidelines and risk of bias assessment figures were generated using the risk of bias visualization tool (Robvis) [13,14]. The revised tool for risk of bias in randomized trials (RoB2) was used for randomized trials (RCT); the revised tool for risk of bias in non-randomized study of intervention (ROBINS-I) was used to assess potential bias in non-randomized studies (NRS) (Figure 4 and Figure 5).

## 3. Results

A total of 602 articles were identified using the key words detailed in this study. Of these, 580 were excluded and 22 were selected as appropriate for this SR, after undergoing screening for eligibility (Figure 6). Clinical evidence on the relationship between plasmatic Hcy concentration and ABI in neurocritical care patients are presented in the following three categories: Hcy in acute ischemic stroke (AIS), Hcy in traumatic brain injury (TBI) and Hcy in intracranial hemorrhage (ICH)/subarachnoid hemorrhage (SAH).

### 3.1. Hcy in AIS

A total of six papers on Hcy and AIS were retrieved, as follows: three RCTs and three prospective NRS. A total of 11,784 patients were analyzed. Data will be displayed in a chronological sequence, according to publication. According to the selected evidence, HHcy predicts outcomes, including neurological deterioration and increased mortality, in AIS patients [15,16,17,18,19,20].

In 2014, the analysis of 396 randomized patients with AIS revealed that early neurological deterioration (END) occurred within 7 days after the admission of individuals with high levels of Hcy. Specifically, plasmatic levels above 10.3 μmol/L of serum Hcy were independent predictors for END, defined as an increase of ≥1 point in motor power or an increase of ≥2 points in the total National Institute of Health Stroke Scale score. This analysis proves that there was an association between HHcy and END in patients with AIS [15].

In 2015, 3799 patients admitted with AIS in Tianjin were included in a prospective clinical trial and categorized according to the following two main stroke subtypes: large-artery atherosclerosis and small-vessel occlusion. Hcy levels were collected within 24 h. In AIS patients with the large-vessel atherosclerosis subtype and elevated Hcy levels (>18.6 μmol/L), death occurred with a 1.61-fold increased risk. However, this correlation was not significant in the small-vessel occlusion subtype, demonstrating that HHcy, in an acute phase of AIS, can predict mortality [16].

In 2017, a prospective multi-center study held in China, data from 3309 AIS patients were analyzed to determine whether the association between hypertension and stroke could be explained through an effect on plasmatic Hcy. Baseline Hcy concentrations were quantitatively determined via an enzymatic cycling assay. The primary outcome was a combination of death and major disability at 3 months after hospitalization. Higher plasmatic Hcy concentrations were associated with an increased risk of the primary outcome in women, but not in men, and further studies are necessary to replicate these findings and to clarify the potential sex-specific mechanisms [17].

In 2018, authors analyzed 598 patients affected by AIS and elucidated that high plasmatic levels of Hcy and high sensitivity of C-reactive protein (hs-CRP)—above 4.65 μmol/L and 1.90 mg/L, respectively—were associated with post-stroke depression (PSD): a frequent mood disorder that occurs after a stroke, diagnosed 3 months after the acute phase by the use of the 24-item Hamilton Depression Rating Scale [18].

In 2019, 638 ischemic stroke patients with elevated blood pressure were recruited in a prospective multi-center study and 12 circulating biomarkers were measured in these participants. Cognitive impairment was assessed at 3 months after the stroke according a Mini-Mental State Examination (MMSE) score <27 or a Montreal Cognitive Assessment (MoCA) score <25. A combination of rheumatoid factor, matrix metalloproteinase-9 and total Hcy, improved the risk prediction of cognitive impairment in AIS patients with elevated blood pressure, demonstrating that Hcy is involved in several pathophysiological pathways that cause post-stroke cognitive impairment [19].

In 2020, 3044 patients with acute minor ischemic stroke or high-risk transient ischemic attack were randomized into the following two groups: clopidogrel plus aspirin or aspirin alone. The models used to assess the interaction of Hcy levels with randomized antiplatelet therapy on efficacy and safety outcomes were Cox proportional hazards models. A significant interaction was found between Hcy and the randomized antiplatelet therapies on recurrent strokes in women, but not in men. Comparing clopidogrel plus aspirin with aspirin alone, the first group displayed a significantly decreased risk of recurrent stroke in women without elevated Hcy levels [20].

In conclusion, according to the literature, HHcy is a valid biochemical indicator of poor cognitive performances and poor prognosis, in terms of mortality, in patients admitted for neurocritical care with AIS. Moreover, in female patients with minor ischemic stroke or high-risk transient ischemic attack, Hcy could be used to discriminate the effects of dual and single antiplatelet therapy.

### 3.2. Hcy in TBI

A total of three papers on Hcy and TBI were retrieved, as follows: one controlled clinical trial (CCT), one cohort study, and one cross-sectional study. A total of 1035 patients were analyzed. Data will be displayed according to the chronological order of the collected evidence [21,22,23]. According to the selected evidence, HHcy is presented as a predisposing factor for clinical complications, including epilepsy, cognition impairment and deteriorated GCS score in patients with TBI. 

In 2011, in a cohort of military personnel, the role of the MTHFR C677T variant gene as a predisposing factor for post-traumatic epilepsy (PTE) was explored. The authors randomly selected 800 PTE patients and 800 matched controls and isolated sufficient genetic material. The probability of post-traumatic epilepsy increased in the subjects with the TT versus CC genotype (crude OR = 1.52 [1.04–2.22], *p* = 0.031; adjusted OR = 1.57 [1.07–2.32], *p* = 0.023), which shows that the MTHFR enzyme is a biologically plausible genetic risk factor for epilepsy, and HHcy can be assumed as a proconvulsant agent [21].

In a prospective case-control study held in 2015, the authors elucidated a significant correlation between plasmatic Hcy levels and prognosis in patients with TBI. Specifically, there was a significant relationship between Hcy plasmatic levels (20.91 μmol/L ± 15.56), the Glasgow Coma Scale (GCS) score at discharge and the Marshall score at 24 h post admission in 150 patients admitted with TBI, versus the control group (7.45 μmol/L ± 13.54, *p* = 0.000). In addition, there was a significant difference in mean plasmatic Hcy levels between patients who died and patients who were still alive at the end of the study period, according to the Glasgow Outcome Scale score [22].

In a cross-sectional study conducted in 2016, a significant correlation between Hcy, endothelial dysfunction and cognitive impairment was pointed out in TBI patients. The plasmatic levels of Hcy, intercellular adhesion molecule-1 (ICAM-1) and vascular adhesion molecule–1 (VCAM-1) were measured in 85 TBI patients, as long as the pulsatility indices (PIs) of the middle cerebral artery by transcranial color-coded Doppler ultrasonography. The cognition status was assessed by Montreal Cognitive Assessment (MCA) and Mini-Mental State Examination (MMSE). The results demonstrated higher levels of Hcy, ICAM-1 and VCAM-1 in patients who died in hospital or during the 6 months after TBI than in survivors; a significant link was found between Hcy and cognition impairment according to MCA and MMSE, and cerebral hemodynamic status according to PI (*p* = 0.000 for all) [23].

In conclusion, the collected evidence reveals that Hcy-induced oxidative stress, overstimulation of NMDA glutamate receptors and microvascular inflammation, all contribute to brain injury, which leads to a significantly disadvantageous outcome. After TBI, a significant link between HHcy and PTE, a deteriorated GCS score, cognition impairment, and death has been elucidated.

### 3.3. Hcy in ICH and SAH 

A total of four papers on Hcy, ICH and SAH were retrieved, as follows: one retrospective study, one cohort study, and two case-control studies. A total of 410 patients with ICH or SAH were analyzed. Data will be displayed according to the chronological order of the evidence collected [24,25,26,27]. HHcy is presented as a predictor of clinical complications, including an increased volume of hematoma, and adverse functional outcomes, including a deteriorated GCS score and death.

In 2015, 69 patients admitted within 24 h after an ICH, were allocated into the following two groups, based on admission plasmatic Hcy levels: a low-Hcy group (≤14.62 μmol/L) versus a high-Hcy group (>14.62 μmol/L). Elevated Hcy levels significantly correlated with a larger hematoma volume in patients with thalamoganglionic ICH, but not in lobar or infratentorial ICH, and poor outcomes were not significantly different between the two groups [24].

In 2018, a study was conducted on 42 patients with spontaneous ICH, with at least one of the following biomarkers recorded on admission: tumor necrosis factor (TNF) alpha, C-reactive protein (CRP), vascular endothelial growth factor (VEGF) and Hcy. The primary outcomes included death, GCS score on admission, END and hemorrhage size. The results elucidated that TNF alpha, CRP, and Hcy levels were not found to predict mortality, and did not correlate to GCS on admission or the initial hemorrhage size [25].

Another study enrolled 150 subjects in a prospective trial, distributed as follows: 100 SAH patients and 50 healthy controls. Plasmatic Hcy levels were determined and MTHFR polymorphism (C677T, A1298C) was screened. Hcy levels were found to be significantly higher in the SAH patients than in the healthy control adults. The frequency of the MTHFR C677T genotype—CT and TT—was significantly higher in the SAH group when compared to the healthy controls. The study suggests that a higher frequency of the MTHFR C677T allele may contribute to the etiopathology of SAH through an increase in Hcy levels [26]. In another study published in 2018, the role of the cystathionine β-synthase (CBS) enzyme and Hcy in SAH was investigated. The authors prospectively enrolled 149 SAH patients and 50 controls. Common CBS polymorphisms were detected using genotyping assays, and an analysis of the associations between CBS polymorphisms and SAH was performed. The GG genotype of the CBS G/A single-nucleotide polymorphism (rs234706) was independently associated with an adverse functional outcome (modified Rankin Scale score of 3–6) at discharge and last follow-up, demonstrating how Hcy metabolism and enzymatic pathways are implicated in SAH prognosis [27].

In conclusion, even if different papers reveal a link between HHcy and increased hematoma size in ICH, further evidence is necessary to validate the hypothesis of worsened clinical outcomes and increased mortality in these patients. In SAH patients, an adverse functional outcome has been elucidated in the case of HHcy.

## 4. Discussion

This SR originally reports clinical evidence on the relationship between plasmatic Hcy concentration and brain-injured neurocritical care patients. The collected evidence reported in this SR suggests that Hhcy is not only an independent risk factor for ABI, but also a marker of poor prognosis in the cases of stroke, TBI, ICH and SAH. In patients with AIS, Hcy plasmatic values ≥14 μmol/L are correlated with recurrent cerebrovascular events, white matter hyperintensity in periventricular and frontal areas and an increased risk of being discharged with a poor functional status [15,16,17,18,19,20]. After TBI, a significant link between HHcy, PTE, deteriorated GCS, cognition impairment and death has been elucidated [21,22,23]. In patients with ICH, further evidence is necessary to validate the hypothesis of worsened clinical outcomes and increased mortality, but, in SAH patients admitted with HHcy, an adverse functional outcome has been elucidated [24,25,26,27].

Similar results, with a predictive role of HHcy, have been highlighted in patients affected by coronary heart disease (CHD). In 1132 patients recruited between 2014 and 2016, who received stent implantation after CHD, the authors demonstrated how post-interventional HHcy was an independent risk factor for in-stent restenosis, with a worsened prognosis [28]. Similarly, in 93 patients presenting with venous thromboembolism, the plasmatic values of Hcy were higher than in healthy controls [29].

A possible mechanism, explaining the increased cardiac and cerebrovascular complications in patients presenting with HHcy, can refer to a common genetic variation in the methylene tetrahydrofolate reductase gene (MTHFR) *C677T*. In particular, Hcy levels were higher (over 20%) in patients with the “TT” genotype than in patients with the “CC” genotype of the *C677T* polymorphism [30,31,32], leading to significant vascular endothelial dysfunction. Other evidence has shown that the genetic polymorphism of an enzyme involved in the conversion of Hcy to Cys (cystathionine β-synthase (CBS) *T833C*) is associated with severe forms of HHcy and an increased risk for subacute thrombosis and subsequent stroke [33,34,35]. In TBI, the MTHFR enzyme is a biologically plausible genetic risk factor for PTE; HHcy can be assumed as a proconvulsant agent and is associated with a worse GCS score after trauma. In patients admitted with ICH and SAH, an increase in hemorrhage size and mortality, respectively, has been proven in the case of HHcy as a consequence of chronic vessel injury [24,25,26,27].

Different clinical trials tested folic acid and vitamin B for the prevention and treatment of HHcy-related increased risk in neurocritical care, where the effects on blood pressure management and deep venous thrombosis (DVT) prevention may play a significant role in reducing mortality. In the body, Hyc can be recycled into methionine or converted into cysteine, with the aid of vitamin B. Normal plasmatic levels of Hyc are maintained by remethylation of Hyc to Met by the N5-methyl tetrahydrofolate enzyme, in the presence of folate and vitamin B12 (2). Vitamin B6, instead, is essential for the catabolism of Hys to cysteine; serine hydroxymethyltransferase is a B6-dependent enzyme [30,31,32,33,34,35] (Figure 1). In an RCT published in 2017, 90 patients with HHcy and stroke were randomly divided into the following two groups: the treatment group was administered folic acid and vitamin B12, while the non-treatment group was not. In the treatment group, the recurrence rate of lower-limb deep static vein thrombosis was 4.4%, which was lower than in the non-treatment group (28.9%, *p* < 0.05). The mechanism of action may be to prevent the recurrence of DVT by reducing the levels of Hcy [36]. In another RCT published in 2018, 224 cases of AIS and H-type hypertension (HHcy and hypertension) were selected and randomly divided into the following two groups: the control group was treated with conventional therapy, while the observation group was treated with 500 µg of mecobalamin three times a day, in addition to conventional therapy. Mecobalamin (methyl-vitamin B12) belongs to the vitamin B group and is one of the active analogous structures of vitamin B12, the essential cofactor for Met synthase. The authors compared the plasmatic Hcy–CRP levels, carotid plaques and NIH stroke scale/scores (NIHSS) between the two groups, with the following promising results: mecobalamin reduced plasmatic Hcy, the levels of plasma inflammatory factors and the volume of carotid artery plaques, resulting in a more significant functional recovery [37].

The literature demonstrates an interaction between antiplatelet therapy and the effects of folic acid-based Hcy-lowering therapy on major vascular events. In a post hoc subanalysis of the VITATOPS trial, 8164 patients with recent stroke or transient ischemic attack were randomly allocated to a double-blind treatment with either a placebo or B vitamins (2 mg folic acid, 25 mg vitamin B_6_, and 500 μg vitamin B_12_), and were followed up for a median of 3–4 years for the primary composite outcome of a stroke, MI or death, due to vascular causes. Among the participants taking antiplatelet drugs at baseline, B vitamins had no significant effect on the primary outcome. In contrast, among the participants not taking antiplatelet drugs at baseline, B vitamins had a significant effect on the primary outcome. These results evidence that B vitamins might play a role in the prevention of ischemic events in high-risk individuals with an allergy, intolerance or lack of indication for antiplatelet therapy [38].

Among the limitations of this SR, we mention that there was a limited number of high-quality studies; out of the 22 studies presented, only a few were randomized or controlled. Only three studies were included in the chapter “Hcy and TBI”, due to a high volume of animal research on the topic, and only four in the chapter “Hcy and ICH and SAH”. Another limitation of this SR is the breadth of the patients included. Despite the intention to analyze each study, including the unfavorable prognosis, unique results could not be achieved because the studies were too different. Further evaluation of the evidence will be necessary in the future to better explore this field. In spite of this, all the included studies have a low or moderate risk of bias. A serious risk of bias was detected in only two papers (Figure 4 and Figure 5).

## 5. Conclusions

Several recent evidences elucidate how Hcy plasmatic levels influence a patient’s prognosis in ABI and, in some cases, the risk of recurrence, appearing as a valid biochemical marker for risk stratification in brain-injured neurocritical care patients. Further RCTs are necessary to test the effects of folate, B6 and B12 supplementation in the primary and secondary prevention of cerebrovascular events. However, a nutraceutical approach, including folic acid and the vitamins B6 and B12, appears to reduce the risk of thrombosis, and cardiovascular and neurological dysfunction in patients with severe HHcy, who are admitted to critical care. In the meantime, especially when ideal Hcy levels are far from being reached, a nutraceutical approach appears to be safe, well-tolerated and effective in reducing plasmatic Hcy.

## Figures and Tables

**Figure 1 jcm-11-00394-f001:**
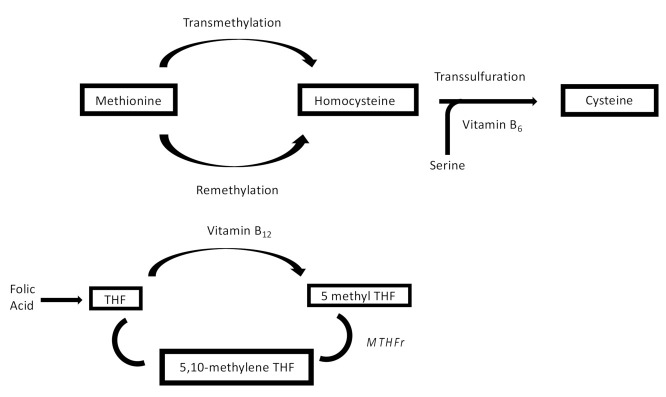
Role of Hcy in folate metabolism. THF: tetrahydrofolate; MTHFR: 5,10-methylene tetrahydrofolate reductase (gently taken from kidney-international.org).

**Figure 2 jcm-11-00394-f002:**
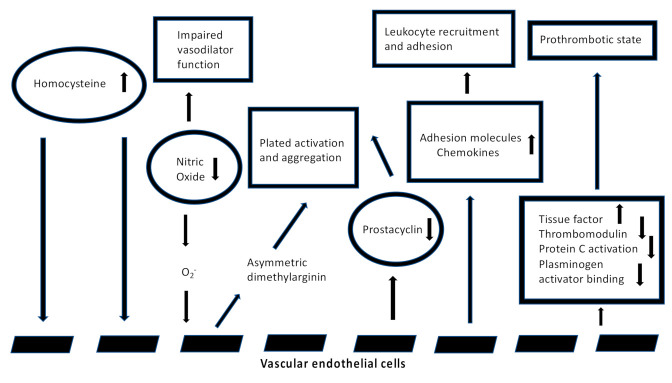
Mechanism of endothelial damage by HHcy (modified by Weiss, N. et al., *Vascular Medicine* 2002, *7*, 227–239).

**Figure 3 jcm-11-00394-f003:**
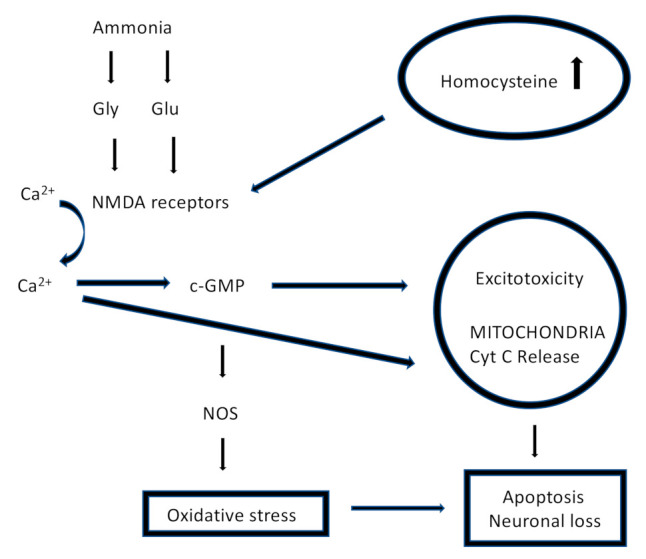
Mechanism of neurotoxicity mediated by Hcy (modified by Choudhury S. et al., Medical Hypotheses Volume 85, Elselvier, Issue 1, 2015; pp. 64–67). Gly: glycine; Glu: glutamate; NMDA receptors: N-methyl-D-aspartate receptors; NOS: nitric oxide synthase.

**Figure 4 jcm-11-00394-f004:**
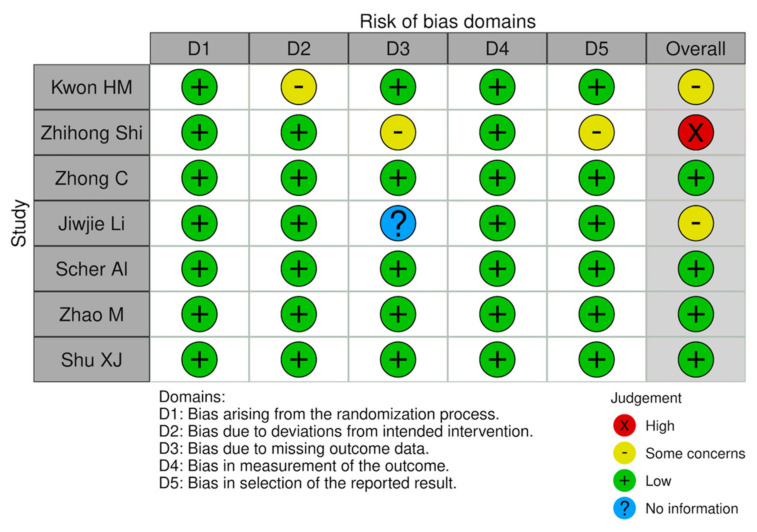
Risk of bias plot of RCTs included in this SR (RoB2).

**Figure 5 jcm-11-00394-f005:**
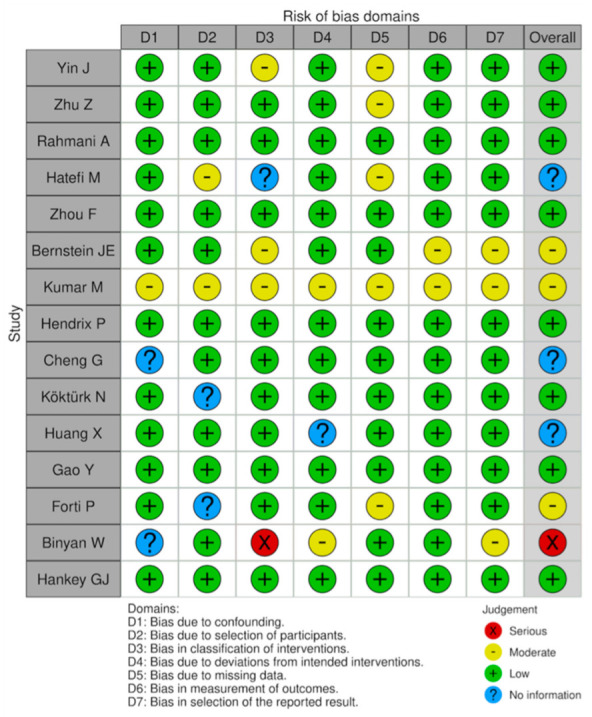
Risk of bias plot of NRS included in this SR (ROBINS-I).

**Figure 6 jcm-11-00394-f006:**
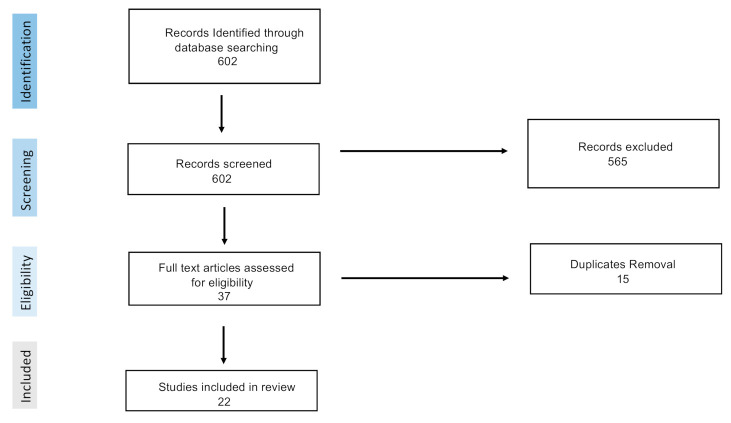
Flow chart of identification of clinical trials included in this systematic review.

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
