# Peer review of "Homocysteine Plasmatic Concentration in Brain-Injured Neurocritical Care Patients: Systematic Review of Clinical Evidence"

_jcm, 2022, doi:10.3390/jcm11020394_

Round 1

Reviewer 1 Report

Present manuscript provides systematic review about impact of homocysteinemia in brain injured neurocritical-care patients. There are some minor problems that have to be fixed:

Some expressions are confusing. For example, Although Hcy is not directly involved 37 in protein synthesis, its role in folate metabolism and choline catabolism is fundamental 38 to regulate Met activity and for the synthesis of several proteins: its ability to donate me-39 thyl groups is essential for the synthesis of methylated compounds, while the inorganic 40 sulfate is pivotal for the synthesis of sulfur-containing amino acids.

Fig. 1 is incomplete. Serine as acceptor of sulfur is missing in transsulfuration reaction.

Fig. 3 Typo error - Mitocondria

Discussion describing impact of vitamins B6 and B12 as well as the impact of folic acid is not well arranged and should be modified.

Author Response

Thank you for the availability in the correction and for the work done on our article. These corrections are precious for each author. 
See the attachment below to read our corrections. 
Regreats, have a nice day. 

F.Bilotta

Reviewer 2 Report

The authors aimed to elucidate the relationship between plasmatic homocysteine concentratin and acute brain injury in neurocritical-care patients. A systematic review was conducted on 22 existing literatures on acute ischemic stroke, traumatic brain injury, intracranial hemorrhage, and subarachnoid hemorrhage. However, the scope of 'neurocritical-care patients' included in the study is too broad. In addition, the end points used to indicate poor prognosis in each study are so heterogeneous that it is difficult to draw a unified result by comparing the results of each study.

Author Response

Thank you for read our manuscript and correct it. We are grateful to try to collab with this journal and we really hope to do it. We understand this problem and we know it yet. Please, see the attachment below. Regards, have a nice day.    F. Bilotta

Round 2

Reviewer 2 Report

Thanks for the revision.